# Evaluating the Safety Risk of Rural Roadsides Using a Bayesian Network Method

**DOI:** 10.3390/ijerph16071166

**Published:** 2019-04-01

**Authors:** Tianpei Tang, Senlai Zhu, Yuntao Guo, Xizhao Zhou, Yang Cao

**Affiliations:** 1School of Transportation and Civil Engineering, Nantong University, Nantong 226019, China; caoyangnt@ntu.edu.cn; 2Business School, University of Shanghai for Science and Technology, Shanghai 200093, China; xizhaozhou@163.com; 3Lyles School of Civil Engineering, Purdue University, West Lafayette, IN 47907, USA; guo187@purdue.edu

**Keywords:** traffic safety, Bayesian Network, run-off-road, roadside features, rural roads

## Abstract

Evaluating the safety risk of rural roadsides is critical for achieving reasonable allocation of a limited budget and avoiding excessive installation of safety facilities. To assess the safety risk of rural roadsides when the crash data are unavailable or missing, this study proposed a Bayesian Network (BN) method that uses the experts’ judgments on the conditional probability of different safety risk factors to evaluate the safety risk of rural roadsides. Eight factors were considered, including seven factors identified in the literature and a new factor named access point density. To validate the effectiveness of the proposed method, a case study was conducted using 19.42 km long road networks in the rural area of Nantong, China. By comparing the results of the proposed method and run-off-road (ROR) crash data from 2015–2016 in the study area, the road segments with higher safety risk levels identified by the proposed method were found to be statistically significantly correlated with higher crash severity based on the crash data. In addition, by comparing the respective results evaluated by eight factors and seven factors (a new factor removed), we also found that access point density significantly contributed to the safety risk of rural roadsides. These results show that the proposed method can be considered as a low-cost solution to evaluating the safety risk of rural roadsides with relatively high accuracy, especially for areas with large rural road networks and incomplete ROR crash data due to budget limitation, human errors, negligence, or inconsistent crash recordings.

## 1. Introduction

Road safety is one of the most important tasks of traffic engineers and is still a big issue for the whole world. On average, there are approximately 1.2 million fatalities and five million injuries in road crashes every year (WHO, 2015). According to the Federal Highway Administration (FHWA) database, roadway departure (RwD) crashes involving run-off-road (ROR) and cross-median/centerline head-on collisions account for approximately 56% of all types of traffic crashes and represent one of the more severe types of crashes [1]. Some recent studies show that ROR crashes on rural roads account for 80% of all ROR fatalities, and around 90% of these crashes occurred on rural two-lane roads [2]. To prevent ROR crashes or reduce the safety risk of rural roadsides, there is a critical need to identify key factors that affect and evaluate roadside safety risk. Given that the budget for rural road safety management is often limited, especially in developing countries, it is necessary to evaluate and classify the safety risk levels of rural roadsides to prioritize budget allocation and possible safety actions.

Although rural road pavement and traffic conditions are steadily improving, and their traffic volume and average operating speed are increasing as the cities expand, the safety facilities of rural roads, such as roadside safety countermeasures, are inadequate and sometimes even missing compared to urban roads, which contributes to the growing number of ROR crashes in recent years [1,2,3]. To address this, some recent studies proposed different methods to evaluate the rural roadside safety risk by analyzing a variety of ROR crash data [4,5,6,7,8,9,10,11,12,13,14]. However, a significant amount data of ROR crashes used in these methods (e.g., crash causation, crash type, crash location, crash severity, traffic condition, etc.) are not available or missing in many countries’ Department of Transportation (DOT) databases (e.g., China) due to human errors, negligence, or lack of training in people in charge of crash recordings. The absence of ROR crash data makes it even more difficult to evaluate the rural roadside safety risk. In addition, some of the safety risk factors for rural roadsides were not included in previous studies, which may not provide a complete picture of rural roadside safety risk. Thus, the relationship obtained in previous studies between the factors and safety risk is possibly also biased [15,16,17,18,19,20,21,22,23,24].

To evaluate the safety risk of rural roadsides for areas with incomplete ROR crash data, this paper proposes a Bayesian Network (BN) based method to evaluate the rural roadside safety risk based on the estimated probability of ROR crash using experts’ judgment. To provide a relatively complete picture of rural roadside safety risk, seven safety risk factors are considered based on their frequency and importance in the literature. A new factor named access point density is introduced to capture some roadside features that were not included in previous studies and may have significant impacts on the frequency and severity of ROR crashes. Then, a BN model is constructed to evaluate the safety risk of rural roadsides and classify them in one of the five safety risk levels. 

To validate the effectiveness of the proposed methods, a case study was conducted using a rural area (6 km^2^) with 19.42 km roads in Nantong, China. Nantong is a prefecture-level city in Eastern China, and its road system contains mainly rural roads (i.e., 80.5% of its 14,694 km roads is classified as rural roads) [3]. 

In summary, the contributions of this paper are:

(1) A Bayesian Network-based method to evaluate the safety risk of rural roadsides for areas with incomplete ROR crash data is proposed.

(2) Access point density is introduced as a relatively new risk factor to evaluate the safety risk of rural roadsides, and the results demonstrated that it has a significant impact on the safety risk of rural roadsides.

(3) This proposed method can be applied to assess the safety risk of road segments or intersections in the absence of ROR crash data as long as safety risk factors for road segments or intersections are re-identified.

The remainder of the paper is organized as follows. In Section 2, literature related to rural roadside safety is reviewed. In Section 3, a safety risk evaluation method for rural roadsides is developed based on a BN method, and the safety risk factors are discussed. Then, a case study and discussions are proposed in Section 4. Finally, some conclusions are provided in the last section.

## 2. Literature Review

### 2.1. Identification of Safety Risk Factors

The premise of evaluating the safety risk is to understand and identify safety risk factors that affect the ROR crashes. Stonex (1960) first proposed that three key safety risk factors, including the steep roadside slopes, deep ditches, and non-traversable obstacles on the roadsides, contributed to the ROR crashes severity [15]. Since then, numerous researchers have investigated the safety risk factors associated with various ROR crashes. Field study and mathematical statistical analysis are the two most commonly used methods to identify these safety risk factors. 

Some studies used a field study and experimental data analysis method to study the impact of the factors related to roadside safety risk. Mclaughlin et al. conducted a 100 car naturalistic driving study to collect the experimental data and revealed that roadside geometric features contributed to the probability of ROR events. Specifically, ROR crashes that occurred on road curves accounted for 30% of all ROR events [16]. Lord et al. conducted site visits to study the ROR crashes on two-way two-lane rural roads in the state of Texas and identified three types of contributing factors, including road geometric design features (i.e., roadside design, shoulder width and type, horizontal curvature, and traffic volume), human factors, and other factors [2]. Fitzpatrick et al. studied the impact of clear zone width and roadside vegetation on driver behaviors by a field data collection. The results demonstrated that drivers had a tendency to drive more closely to the shoulder edge [17]. Another study conducted by the Iowa DOT demonstrated that a 38% reduction in ROR crashes could be achieved after removing/relocating/shielding hazardous objects in the clear zone area by a field study and experimental data analysis [18]. 

To deeply understand the relationship between the factors and the ROR crash severity, some researchers quantified the relationship using logit models and conditional probability models. Lee and Mannering identified the significant contributing factors to the ROR crashes severity using zero-inflated models and nested logit models. They indicated that reducing the number of roadside trees, avoiding cut side slopes, and increasing the distance between shoulder edge and hazardous objects (e.g., trees and utility poles) could reduce the ROR crashes severity [19]. Holdridge et al. developed multivariate nested logit models to identify the most common factors contributing to the severity of fixed-object crashes. The study results highlighted the main factors—the existence of bridge rails, leading ends of guardrails, and wooden poles along the roadsides—that increased the likelihood of severe crashes [20]. Eustace et al. developed an approach to identify the most common factors contributing to ROR events based on generalized ordered logit regression. The authors indicated that road geometric features (e.g., curves), roadway features (e.g., grade), driver conditions, and gender were the key factors contributing to severe ROR crashes [21]. Based on a sufficient number of ROR crash data collected on freeway roads in Portugal, Roque et al. used the multinomial and mixed logit regression models and identified two roadside elements consisting of slops and horizontal curves that have more significant impacts on fatal ROR crashes [22]. Liu and Subramanian identified the factors contributing to single vehicle ROR crashes—horizontal road alignment, roadway geometric features, speed limit, and lighting conditions—by logistic regression based on the crash data from the Fatality Analysis Reporting System [23]. Roque and Jalayer developed a hazard-based duration model to understand the distance traveled by errant vehicles in ROR crashes and their associated factors. Based on annual ROR crashes data, the extent severity was related to roadway and roadside geometric design features, including lane width and clear zones [24].

Several studies developed evaluation methods based on roadside safety risk factors to identify the main factors impacting the roadside safety risk, including horizontal curve radius, side slop, side slope height, clear zone width, longitudinal grade, distance between non-traversable obstacles and roadway edge, roadside objects density, and sight distance, among others [4,5,6,7,8,9,10,11,12,13,14].

In addition, few studies had the perceived access point density as a key factor contributing to ROR crashes. Results from previous studies showed that access point density was positively correlated with crash frequency and severity [25,26]. Therefore, this paper added access point density to the set of factors and studied its impact on rural roadside safety risk. 

### 2.2. Evaluation Methods for Safety Risk

In an attempt to determine which roadside safety countermeasure to adopt, many studies started with evaluating the safety risk of rural roadsides. Some researchers developed a multi-index comprehensive evaluation model to assess the roadside safety risk. Zegeer et al. proposed a visual and subjective measure, which is a roadside hazard rating (RHR) system, to quantify the roadside hazard level on a scale from one to seven (one being the best) [4]. You et al. proposed a roadside dangerous index based on the probability of run-off-road, exposure of the vehicle to hazardous roadside, and risk severity to evaluate roadside safety. [5]. Loprencipe et al. proposed a numeric index (hazard index) to quantify the overall risk assessment of roadsides based on general characteristics of the road and defined six classes of roadside safety risk [6]. Several studies evaluated the roadside safety risk using mathematical statistical analysis. Li et al. developed a risk assessment model based on the grey cluster method [7]. Pardillo-Mayora et al. developed a roadside hazardousness index (RHI) for two-lane roads to assess the RwD crash severity levels using cluster analysis [8]. Esawey and Sayed proposed a safety performance function (SPF) to evaluate the safety risk related to utility pole crashes using negative binomial regression [9]. Park and Abdel-Aty assessed the safety effectiveness of rural roadsides by estimating crash modification factors (CMFs) using the cross-sectional method based on the crash data in Florida [10]. 

In some literature, some of them used a fuzzy synthetic method. Wei and Zhang established the evaluation index system and developed a set pair analysis model (SPAM) for roadside risk rating assessment [11]. Fang et al. presented an assessment model of roadside environment objective safety based on the probability of run-into-roadside and the roadside objective characteristics using fuzzy judgment [12]. The others used the probability theory. Ayati et al. used an evidential reasoning (ER) method to define a roadside hazard severity indicator and developed crash severity prediction models to evaluate the roadside hazard severity levels [13]. Jalayer and Zhou obtained ROR crash data for a five year time period (2009–2013) from Illinois DOT and used reliability analysis to gauge the RHRs for rural two-lane roads [14]. Table 1 summarizes the main evaluation methods for roadside safety risk in previous studies. 

Apart from the aforementioned methods, the Bayesian Network has been used in traffic safety evaluation domains. Several studies used BN to assess traffic safety, traffic accidents, and traffic accident injury severity [27,28,29]. Some other studies identified the accident black spots and predicted real-time crashes using a BN method [30,31]. However, few studies used BN to assess rural roadside safety risk. Additionally, some studies investigated the application of mathematical statistical analysis, probability theory, and fuzzy synthetic method to analyze the traffic accidents. Chen et al. used mixed logit models to analyze the hourly crash likelihood of highway segments and investigate the injury severities of truck drivers in single-and multi-vehicle accidents on rural highways [32,33]. Ma et al. developed multivariate space-time models to jointly analyze crash frequency by injury severity levels in fine temporal scale [34]. Wen et al. studied crash frequency on freeway segments using a Poisson-based count regression with consideration of the spatial effects [35]. Shi et al. proposed a cask evaluation model based on fuzzy and cask theory to assess safety in Chinese rural roads [36]. 

To sum up, the methodology process of evaluating roadside safety performance includes the multi-index comprehensive evaluation method, the mathematical statistical analysis, the fuzzy synthetic method, and the probability theory. The data of ROR crash frequency and severity are usually used to characterize the extent of safety risk. This means that the effectiveness and practicality of these methods are entirely dependent on the data quality of the ROR crash. In reality, many types of ROR crash data are not available or missing in many cases. The study proposes a BN-based method to address this issue. The studied results demonstrate that the proposed approach based on BN has an ability to effectively evaluate the safety risk of rural roadsides in the absence of ROR crash data.

## 3. Methods

### 3.1. BN-Based Evaluation Method

The data for ROR crashes on rural roads are insufficient and difficult to obtain in many countries. In the absence of ROR crash data, it is difficult to develop a road safety risk assessment method. In this section, we propose a Bayesian Network-based method to assess and classify the rural roadside safety risk based on the estimated probability of ROR crashes. 

Due to advantages including bi-directional induction, incorporation of missing variables, and probabilistic inference, BN has gained widespread attention as a method for risk/reliability assessment in fields of industrial engineering, information technology, and social science [37,38,39,40,41,42]. In the traffic safety domain, many studies have used BN to analyze, assess, and predict traffic safety risk and traffic accidents [27,28,29,30,31]. 

To evaluate the rural roadside safety risk, the proposed BN model is shown in Figure 1.

The proposed BN model is composed of parent variable $B_{i}$ and child variable $E_{ij}$. Variable $B_{i}$ represents the decision factor-i (i=1,2,⋯,k, where k is the number of decision factors). Variable $B_{i}$ is aggregated to the objective variable R, which represents the rural roadside safety risk. To ensure relative objectivity and reliability of the experts’ judgment, some experts are invited and divided into n expert panels. Variable $E_{ij}$ represents the expert panel-j (j=1,2,⋯,n, where j is the number of expert panels), and the evidence judged by the expert panels is added into the BN model. A set of directed edges connect the parent variables and child variables. 

In this study, the safety risk in the respective decision factors is represented by the probabilities that a road segment falls in Hi (high safety risk category) in the respective decision factors. These probabilities are calculated using BN based on the experts’ knowledge and experience. $P\left(B_{i}=Hi\right)$ denotes the probabilities that a road segment falls in Hi in decision factor-i. Each $P\left(B_{i}=Hi\right)$ is calculated and updated using evidence judged by different expert panels.

In the BN model, each variable has a finite set of mutually exclusive states. The parent variable $B_{i}$ indicates the situation that a road segment falls in Hi (high safety risk category) or Lo (low safety risk category) in decision factor-i. The child variable $E_{ij}$ indicates the situation that a road segment falls in a decision criterion, such as $e_{1},e_{2},\cdots ,e_{m}$ (where m is the number of decision criterion in decision factor-i), according to the judgment of different expert panels. A system of decision criteria is formed for each expert panel before their judgment to ensure the consistency of the decision criteria. Different expert panels have different decision criteria, and these uncertainties can be adequately handled by the BN model.

For each $E_{ij}$ with $B_{i}$, there is attached a conditional probability table $P\left(E_{ij}|B_{i}\right)$. If $B_{i}$ has no parents, the table converts to unconditional probabilities $P\left(B_{i}\right)$. The interest of our attention is the updated $P\left(B_{i}\right)$, not the prior $P\left(B_{i}\right)$, thus we assume that $P\left(B_{i}=Hi\right)=\left(B_{i}=Lo\right)=1/2$. Evaluating $P\left(E_{ij}|B_{i}\right)$ is difficult for the experts because variable $E_{ij}$ has a different number of states. Nevertheless, $P\left(B_{i}|E_{ij}\right)$ can be obtained easily because variable $B_{i}$ has only two states of Hi or Lo. Therefore, we can calculate $P\left(E_{ij}|B_{i}\right)$ using Bayesian theorem with $P\left(B_{i}|E_{ij}\right)$, which are evaluated by experts. Note that evaluating $P\left(B_{i}|E_{ij}\right)$ as point values is difficult for experts, thus $P\left(B_{i}|E_{ij}\right)$ are evaluated as band values, and the medians of the band values are employed as the representative values to calculate $P\left(B_{i}|E_{ij}\right)$, as shown in Table 2. If the median of the band value is not appropriate according to the judgment of experts, experts can replace an appropriate one for the median. In summary, $P\left(E_{ij}|B_{i}\right)$ can be calculated using Equation (1):(1)$$\begin{array}{cc} P\left(E_{ij}|B_{i}\right) & =\frac{P\left(B_{i}|E_{ij}\right)P\left(E_{ij}\right)}{P\left(B_{i}\right)}\\ & =\frac{P\left(B_{i}|E_{ij}\right)P\left(E_{ij}\right)}{P\left(B_{i}|E_{ij}=e_{1}\right)P\left(E_{ij}=e_{1}\right)+\cdots +P\left(B_{i}|E_{ij}=e_{m}\right)P\left(E_{ij}=e_{m}\right)}\\ & =\frac{P\left(B_{i}|E_{ij}\right)P\left(E_{ij}\right)}{\sum_{k=1}^{m}P\left(B_{i}|E_{ij}=e_{k}\right)P\left(E_{ij}=e_{k}\right)} \end{array}$$

Refer to $P\left(B_{i}=Hi\right)=\left(B_{i}=Lo\right)=1/2$, similarly, we assume that $P\left(E_{ij}=e_{1}\right)=,\cdots ,=P\left(E_{ij}=e_{m}\right)=1/m$, because the variable $E_{ij}$ has no information provided.

Using $P\left(E_{ij}\right)$ and $P\left(B_{i}|E_{ij}\right)$, $P\left(B_{i}\right)$ can be calculated as Equations (2) and (3):(2)$$P\left(B_{i}=Hi\right)=P\left(B_{i}=Hi|E_{i1}=e_{1}\right)P\left(E_{i1}=e_{1}\right)+\cdots +P\left(B_{i}=Hi|E_{i1}=e_{m}\right)P\left(E_{i1}=e_{m}\right)$$
(3)$$P\left(B_{i}=Lo\right)=1-P\left(B_{i}=Hi\right)$$

Using the evidence judged by experts, each i
$P\left(B_{i}\right)$ can be updated as the following steps:

Step 1. If expert panel-1 judges that variable $B_{i}$ falls in decision criteria $e_{1}$, that is, $P^{*}\left(E_{i1}\right)=\left(1,0,0,\cdots \right)$, where evidence on $E_{ij}$ is denoted by $P^{*}\left(E_{ij}\right)$, the probabilities $P^{*}\left(E_{i1}\right)=\left(1,0,0,\cdots \right)$ are employed as evidence to update $P\left(B_{i}=Hi\right)$, as follows:(4)$$\begin{array}{cc} P^{*}\left(B_{i}=Hi\right) & =P\left(B_{i}=Hi|E_{i1}=e_{1}\right)P^{*}\left(E_{i1}=e_{1}\right)+\cdots +P\left(B_{i}=Hi|E_{i1}=e_{m}\right)P^{*}\left(E_{i1}=e_{m}\right)\\ & =P\left(B_{i}=Hi|E_{i1}=e_{1}\right)P^{*}\left(E_{i1}=e_{1}\right) \end{array}$$
where $P\left(B_{i}=Hi\right)$ updated by evidence is denoted by $P^{*}\left(B_{i}=Hi\right)$.

Step 2. If expert panel-2 judges that variable $B_{i}$ falls in decision criteria $e_{2}$, that is, $P^{*}\left(E_{i2}\right)=\left(0,1,0,\cdots \right)$, the probabilities $P^{*}\left(E_{i2}\right)=\left(0,1,0,\cdots \right)$ are employed as evidence to update $P^{*}\left(B_{i}=Hi\right)$, as follows:(5)$$\begin{array}{cc} P^{**}\left(B_{i}=Hi\right) & =P^{*}\left(B_{i}=Hi|E_{i2}=e_{1}\right)P^{*}\left(E_{i2}=e_{1}\right)+\cdots +P^{*}\left(B_{i}=Hi|E_{i2}=e_{m}\right)P^{*}\left(E_{i2}=e_{m}\right)\\ & =P^{*}\left(B_{i}=Hi|E_{i2}=e_{2}\right)P^{*}\left(E_{i2}=e_{2}\right) \end{array}$$
where $P^{*}\left(B_{i}=Hi|E_{i2}=e_{2}\right)$ can be calculated as follows:(6)$$\begin{array}{cc} P^{*}\left(B_{i}=Hi|E_{i2}=e_{2}\right) & =\frac{P\left(E_{i2}=e_{2}|B_{i}=Hi\right)P^{*}\left(B_{i}=Hi\right)}{P\left(E_{i2}=e_{2}\right)}\\ & =\frac{P\left(E_{i2}=e_{2}|B_{i}=Hi\right)P^{*}\left(B_{i}=Hi\right)}{P\left(E_{i2}=e_{2}|B_{i}=Hi\right)P^{*}\left(B_{i}=Hi\right)+P\left(E_{i2}=e_{2}|B_{i}=Lo\right)P^{*}\left(B_{i}=Lo\right)} \end{array}$$

For a selected road segment, $P\left(B_{i}=Hi\right)$ can be updated on the basis of the evidence judged by different expert panels until the evidence judged by expert panel-n is used. Finally, as all evidence is added into the BN model, the more reasonable $P\left(B_{i}=Hi\right)$ is computed. The updated probabilities $P\left(B_{i}=Hi\right)$ are integrated into the total probability P(R=Hi), as follows: (7)$$P\left(R=Hi\right)=\overset{k}{\underset{i=1}{\sum }}\omega_{i}P\left(B_{i}=Hi\right)$$
where $\omega_{i}$ is the weighting values of the respective decision factors. In this paper, the weighting values of decision factors are not the study objective, thus we can assume that $\omega_{i}=1/k$. Further research is required to discuss the weighting values.

Given the values of P(R=Hi), we define the safety risk levels on a scale of one to five (one being the lowest safety risk and five being the highest) (Table 3).

### 3.2. Safety Risk Factors for Rural Roadsides

Table 4 summarizes the main factors contributing to rural roadside safety risk/ROR crashes in previous studies.

As shown in Table 4, the first seven factors are frequently perceived as the significant factors in previous studies, while the remaining six factors have only appeared once, respectively, in one literature. Taking into account the frequency and significance of each factor in previous research and considering the scalability of factor data, we perceive the first seven factors as the key factors in this paper. 

Apart from the first seven factors identified in most literature, few studies have perceived access point density as a key factor contributing to ROR crashes, which we address in this paper. The previous studied results show that access point density is positively correlated with crash frequency and severity [25,26]. Using the definition of access point in Access Management Manual, this paper defines access point as a node that connects the main road to other levels of roads; compared to the main road, their design standards are relatively lower and have fewer constraints, specifically including classified road, farmland road, and residential road (as shown in Figure 2). For example, in Figure 2, there are 20 access points located on the two sides of a 658 m main road. This means that the access point density of this road is 30.4 point/km, which will seriously affect the operation of the main traffic flow. We demonstrate that access point density is indeed relevant to roadside safety risk and affects the reliability of the evaluation results by an efficient and practical approach in Section 4.3. 

In summary, a set of eight safety risk factors is defined, which includes horizontal curves radius $B_{1}$, longitudinal gradient $B_{2}$, distance between roadway edge and non-traversable obstacles $B_{3}$, side slope grade $B_{4}$, side slope height $B_{5}$, access point density $B_{6}$, density of discrete non-traversable obstacles (e.g., trees, utility poles, buildings, etc.), and $B_{7}$, density of continuous non-traversable obstacles (e.g., worn out roadside safety barriers, unprotected drainage channels, etc.) $B_{8}$. The continuous non-traversable obstacles refer to the obstacles when the longitudinal length is more than 3 m, or the longitudinal distance between two adjacent discrete obstacles is less than 5 m [11,12].

## 4. Case Study

### 4.1. Study Area

The case study area is a rural area of 6 km^2^ (3 km by 2 km) in Nantong, China (as shown in Figure 3). There are many rivulets, access points, residential constructions, and non-traversable obstacles (e.g., trees, utility poles, etc.) along the roadsides in the study area. The portion of the rural roads used in this study consists of 19.42 km two-lane roads, including 5.16 third-class roads (3.5 m of lane width) and 14.26 fourth-class roads (3 m of lane width). The others are arterial highways (e.g., provincial trunk highway) or substandard roads (e.g., farmland road, residential road, etc.), which are not within the scope of assessment. Rural roadway and roadside feature data were obtained from the Nantong DOT and Google Earth Pro. The data were processed to assess and classify the safety risk of rural roadsides.

For the sake of brevity, roadside safety risk assessment is elaborated by using a straight segment with a length of 200 m belonging to the network in the study area as an example segment (as shown in Figure 3). With regard to the risk factors of this example segment, the horizontal curves radius is over 60 m, the longitudinal gradient is 2.5%, the distance between roadway edge and obstacles is 0.8 m, the side slope grade is 1:1.5, the side slope height is 1.5 m, the access point density is 11 point/km, the density of discrete non-traversable obstacles is 8 obstacle/km, and the density of continuous non-traversable obstacles is 0.08 km/km.

### 4.2. Roadside Safety Risk Evaluation

To evaluate the rural roadside safety risk of the study area shown in Figure 3, we divided each rural road into a defined segment length of 200 m. If the last segment of rural road was less than 200 m, we also defined it as a segment. Therefore, the whole road network had 99 segments. Each segment data of roadway and roadside features were sourced from Nantong DOT and Google Earth Pro. 

In order to ensure relative objectivity and reliability of expert judgments, we invited 24 experts in traffic safety domain from different institutions, including experts from departments of transportation, research institutes, universities, and engineering consultancies. These experts included nine senior engineers, eight professors, and seven project managers. The invited experts were divided into three panels for the comprehensive assessment. Through discussion within each panel, the decision criteria of each panel were confirmed. Table 5 presents the decision criteria of the respective factors for each expert panel.

Given the decision criteria of the respective factors for three expert panels (as shown in Table 5), a conditional probability table $P\left(E_{ij}|B_{i}\right)$ could be calculated using Equation (1). Taking the distance between roadway edge and obstacles $B_{3}$ as an example (Table 5), the expert panel-1 has three decision criteria for $E_{31},$ namely $e_{1}$, $e_{2}$, and $e_{3}$. When a road segment falls in Hi, the conditional probability of $B_{3}$ falling in the respective decision criteria is: $P\left(E_{31}=e_{1}|B_{3}=Hi\right)=0.5556$, $P\left(E_{31}=e_{2}|B_{3}=Hi\right)=0.3333$, $P\left(E_{31}=e_{3}|B_{3}=Hi\right)=0.1111$. Conversely, when a road segment falls in Lo, the conditional probability of $B_{3}$ falling in the respective decision criteria is: $P\left(E_{31}=e_{1}|B_{3}=Lo\right)=0.2381$, $P\left(E_{31}=e_{2}|B_{3}=Lo\right)=0.3333$, $P\left(E_{31}=e_{3}|B_{3}=Lo\right)=0.4286$. In a similar way, Table 6 lists the conditional probability $P\left(E_{3j}|B_{3}\right)$ for three expert panels.

Taking the example segment as an example (Figure 3), the distance between roadway edge and obstacles $B_{3}$ is 0.8 m. Table 5 shows that $B_{3}$ falls into criteria $e_{1}$ of variable $E_{31}$, criteria $e_{2}$ of variable $E_{32}$, and criteria $e_{2}$ of variable $E_{33}$, respectively. $P^{*}\left(E_{31}\right)=\left(1,0,0\right)$, $P^{*}\left(E_{32}\right)=\left(0,1,0,0\right)$, $P^{*}\left(E_{33}\right)=\left(0,1,0,0,0\right)$ are employed as evidence to update $P\left(B_{3}=Hi\right)$ using Equation (2) and updated steps, $P\left(B_{3}=Hi\right)=0.6779$. Similarly, the updated probability of other variables $B_{i}$ can be calculated in the same manner. Then, using Equation (7), this example segment’s P(R=Hi)=0.3520 and its roadside safety risk are classified as 2-level. The above calculation results were calculated by BN tools GeNie2.0.

Figure 4 shows the roadside safety risk evaluation results calculated by GeNie2.0. The number of segments whose safety risk level belong to the 1–5 level are 50, 28, 10, 8, and 3, respectively. Figure 5 further visualizes the distribution of road segments with different levels of safety risk. As shown in Figure 5, high-risk roadside segments of 4-level and 5-level are mainly concentrated in the dense residential areas and locations with continuous sharp-curves. 

To identify the factors contributing to the relatively high-risk level of rural roadsides, 11 high-risk segments (Figure 5) are selected and defined with the alpha-numerical code $S_{r}$, where r ranges from 1 to 11. $S_{1}$, $S_{4}$, and $S_{8}$ are located in the dense residential areas with many residential access points and discrete non-traversable obstacles, such as utility poles and buildings along the roadsides, and the distance between roadway edge and obstacles is less than 1 m. $S_{2}$ and $S_{3}$ are also concentrated in the dense residential areas with two continuous sharp-curves whose radii are less than 30 m, which may cause vehicle run-off-road crashes in the absence of traffic calming measures to reduce vehicle speed. The side slope grade (side slope height) of $S_{5}$ and $S_{7}$ are 1:1.5 (1.5 m) and 1:1 (1.1 m), respectively, and $S_{7}$ has a long longitudinal slope with a gradient of 2.8%, resulting in a higher risk. $S_{6}$ and $S_{11}$ are due to the continuous non-traversable obstacles, which are unprotected drainage channels along the roadsides. The key high-risk factors for $S_{9}$ are sharp-curve and higher density of residential access points. $S_{10}$ has two continuous sharp-curves with radii less than 25 m and unprotected drainage channels along the roadsides. The evaluated results highlight that the aforementioned factors have a more significant impact on the safety risk of rural roadsides. For these high-risk factors, we can take some safety management measures and install the safety facilities to reduce the safety risk level of these rural roadsides.

### 4.3. Effectiveness of Roadside Safety Risk Evaluation Results

To validate the effectiveness of the evaluation method and results, we obtained the historical ROR crash data for a two year period (2015 to 2016) from the Nantong Department of Traffic Police (DOTP). The locations of crashes are reported as road addresses or intersections, which are geocoded to the rural road network in the study area. Table 7 illustrates the crash severity and crash frequency in the study area.

To make ROR crash data comparable to the safety risk, a crash severity index (CSI) is introduced to represent the crash severity per segment. The CSI can be defined on crash frequency and crash severity, and calculated as follows:(8)$$CSI=\alpha_{fatal}n_{fatal}+\alpha_{injury}n_{injury}+\alpha_{PDO}n_{PDO}+\alpha_{NC}n_{NC}$$
where $\alpha_{fatal}$, $\alpha_{injury}$, $\alpha_{PDO}$, and $\alpha_{NC}$ are the crash severity coefficients of fatal, injury, property damage only, and no crash, respectively. $n_{fatal}$, $n_{injury}$, $n_{PDO}$, and $n_{NC}$ are crash frequency of fatal, injury, property damage only, and no crash, respectively (as shown in Table 7).

Figure 6 shows the distribution of road segments with different levels of CSI calculated by Equation (8). Compared to Figure 5, road segments with a higher safety risk level tend to have a higher value of CSI. According to the comparison of CSI and P(R=Hi), Figure 7 shows that 87.9% of all segments’ CSI and P(R=Hi) values fall into the same risk level, and among the 12 high-risk segments, only one segment’s CSI and P(R=Hi) value is not within the same risk level. Consequently, it is reasonable to assume that the greater the safety risk level is, the higher the CSI value is. As shown in Figure 8, the polynomial regression results between the safety risk levels and CSI have a correlation coefficient of 0.8538, suggesting a high correlation between safety risk levels and CSI. These studied results show that the proposed method can be used to effectively evaluate the safety risk of rural roadsides, and this BN-based approach can cope with the uncertainty of different decision criteria judged by different expert panels.

To validate the importance of factoring the perceived access point density in ROR crashes (which was often not included in previous studies), we used the proposed method to compare the results with the perceived access point density and the results without it. We removed it from the set of factors and used the proposed approach to reevaluate the study area. Note that we defined the results evaluated by a set of eight factors as an experiment group and those by a set of seven factors as a control group. As shown in Figure 9, the comparison results show that only 62.6% of all segments’ CSI and P(R=Hi) values fall into the same risk level in a control group, which is much lower than the proportion in the experiment group. Specifically, among 12 high-risk segments, seven segments have a lower risk level using the proposed method than the risk level using CSI. Through analyzing the roadside features of these seven segments, we find that these segments have a high density of access points, which is a completely ignored risk factor in the control group. Figure 10 illustrates that the safety risk levels are found to increase in consistency with the value of CSI, which is in good agreement with the findings in the experiment group. Nevertheless, the correlation coefficient is 0.4252, which is lower than the experiment group. Furthermore, summary statistics for CSI corresponding to safety risk levels evaluated by the experiment group and control group are shown in Table 8. Table 8 and Figure 11 illustrate that the standard deviation of CSI corresponding to the safety risk levels of the experiment group is significantly lower than that of the control group. Therefore, the evaluation results using a set of factors that includes the access point density can characterize the safety risk of rural roadsides more accurately, which means a risk factor of access point density significantly contributes to the safety risk of rural roadsides.

## 5. Conclusions

Previous studies relied heavily on the quality of ROR crash data to evaluate rural road safety risk, but such data may not be available or complete in many rural areas due to lack of funding or manmade errors. This study proposed a BN-based method using experts’ judgments on the conditional probability of different safety risk factors to evaluate the safety risk of rural roadsides. By summarizing the roadside safety risk factors identified in previous studies and introducing a new factor that may have a significant impact on roadside safety risk, eight factors were considered, including horizontal curves radius, longitudinal gradient, side slope grade, side slope height, distance between roadway edge and non-traversable obstacles, access point density, density of discrete non-traversable obstacles, and density of continuous non-traversable obstacles.

To validate the proposed method, a case study was conducted using a rural area (6 km^2^) with 19.42 km roads in Nantong, China. By comparing the safety risk levels of the proposed method and CSI generated from ROR crash data from 2015–2016, the results show that the proposed method can generate a consistent result with ROR crash data, meaning the higher the safety risk levels are, the higher the CSI is. Additionally, by comparing the evaluating results with the perceived access point density and the results without it, we also found that access point density significantly contributed to the safety risk of rural roadsides. These results demonstrate that the proposed method can cope with inconsistent expert judgments and can serve as a low-cost solution to evaluate the safety risk of rural roadsides, especially for areas with incomplete ROR crash data.

A limitation of this study was that we could only obtain two year (2015–2016) data from Nantong DOTP due to government restrictions on traffic accident data disclosure. Additional studies are needed to include three or more years of data to validate our research once such data are available to the public. Note that the modeling precision relied on the decision criteria judged by the experts and rationality of safety risk factor identification. Therefore, for future research, it is worth strengthening the reliability of expert judgments by simulating the ROR crash scenarios to obtain a dataset of crash frequency and severity and calculate the probabilities of ROR crashes caused by the respective safety risk factors. Another potential research direction is to include additional factors related to the roadside safety risk, such as lane width, sight distance, etc., which can potentially improve the quality of the proposed method in evaluating the safety risk of rural roadsides.

## Figures and Tables

**Figure 1 ijerph-16-01166-f001:**
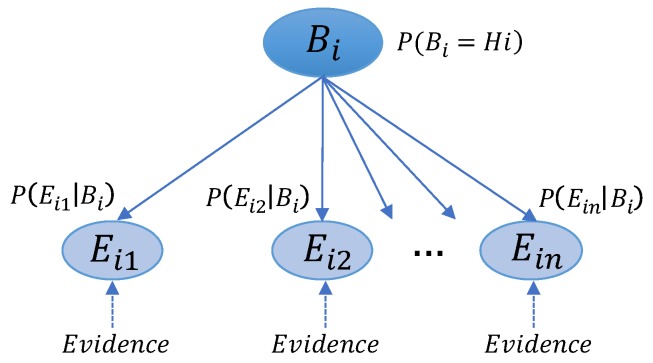
A Bayesian Network (BN) model in decision factor-i.

**Figure 2 ijerph-16-01166-f002:**
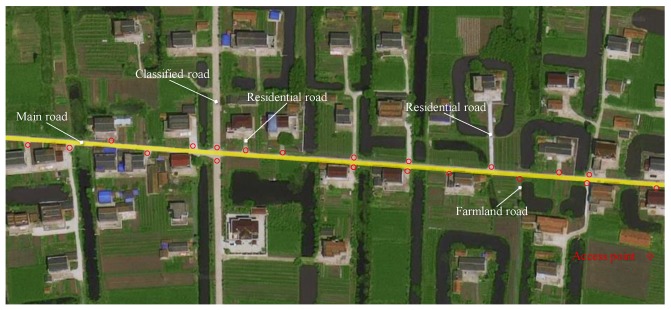
Definition of access point.

**Figure 3 ijerph-16-01166-f003:**
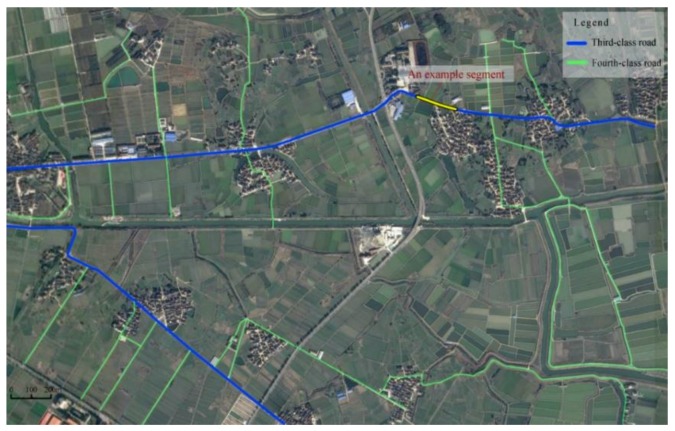
Study area.

**Figure 4 ijerph-16-01166-f004:**
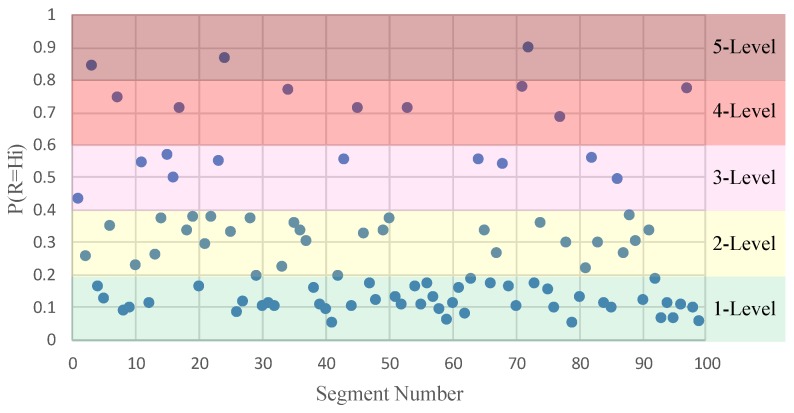
Safety risk levels distribution for each segment.

**Figure 5 ijerph-16-01166-f005:**
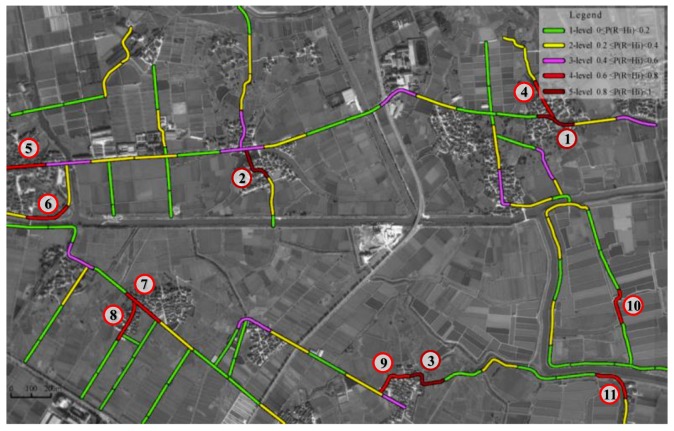
Safety risk levels distribution in the study area.

**Figure 6 ijerph-16-01166-f006:**
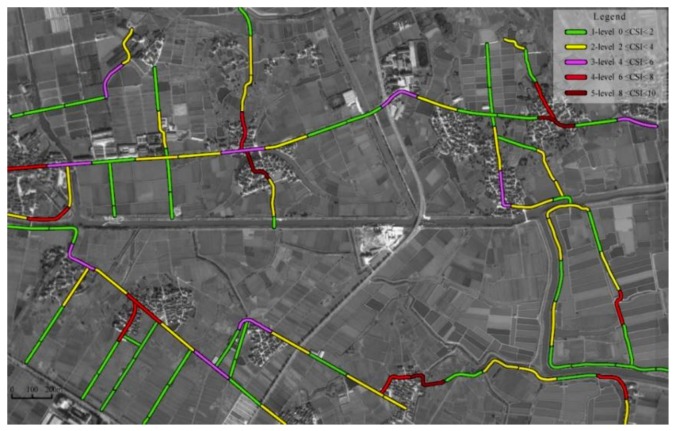
Levels of crash severity index (CSI) distribution in the study area.

**Figure 7 ijerph-16-01166-f007:**
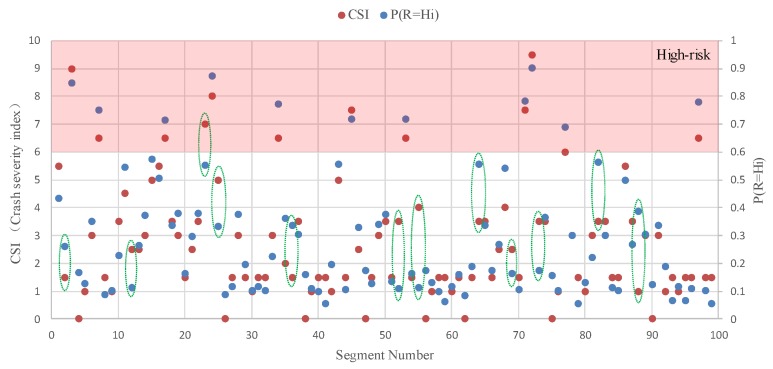
Comparison of CSI and P(R=Hi) (an experiment group).

**Figure 8 ijerph-16-01166-f008:**
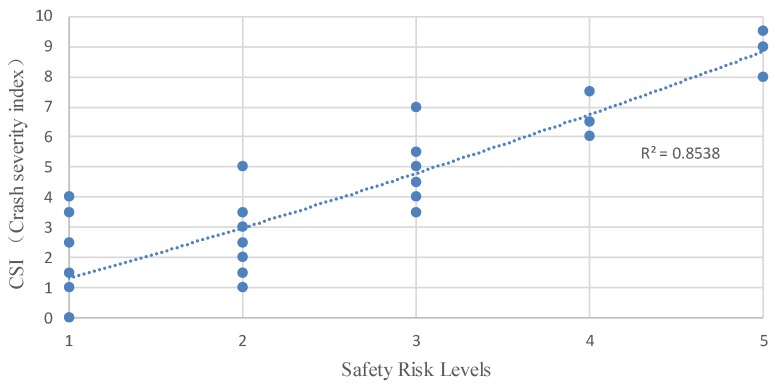
Safety risk levels versus CSI (an experiment group).

**Figure 9 ijerph-16-01166-f009:**
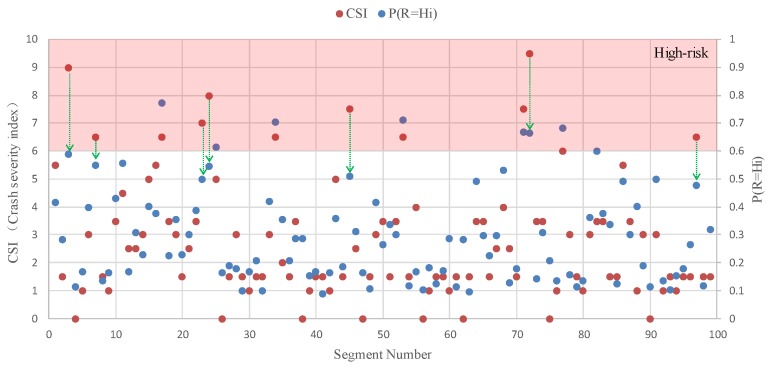
Comparison of CSI and P(R = Hi) (a control group).

**Figure 10 ijerph-16-01166-f010:**
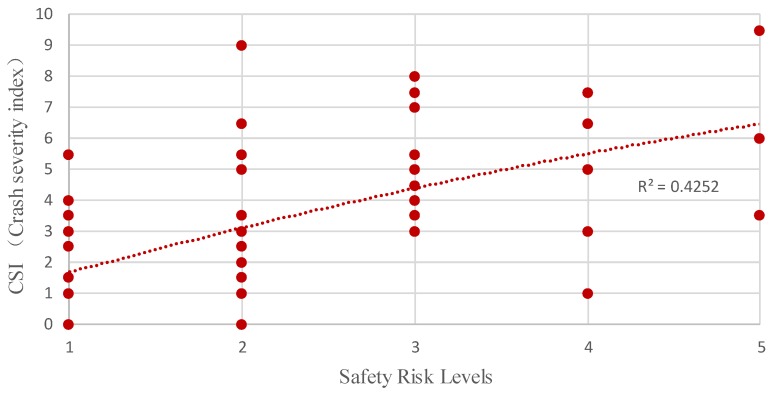
Safety risk levels versus CSI (a control group).

**Figure 11 ijerph-16-01166-f011:**
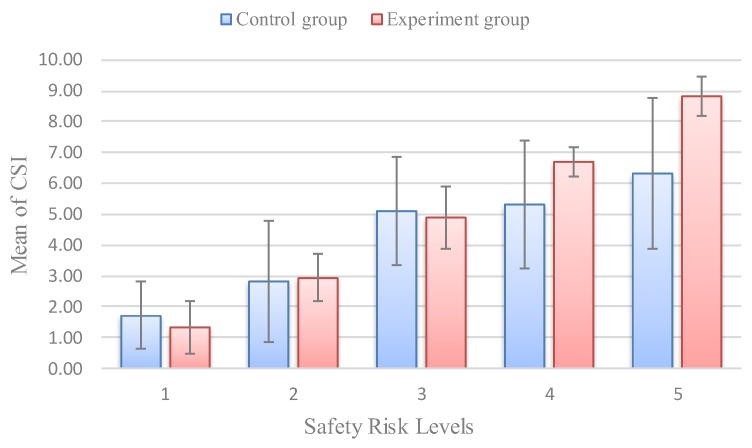
Comparison of mean and SD of CSI corresponding to levels between an experiment group and a control group.

**Table 1 ijerph-16-01166-t001:** Studies on evaluation methods for roadside safety risk.

Method Type	Methods	References
Multi-index comprehensive evaluation method	Roadside hazard rating (RHR) system	Zegeer et al. (1987) [4]
	A roadside dangerous index	You et al. (2010) [5]
	Hazard index	Loprencipe et al. (2018) [6]
Mathematical statistical analysis	Grey cluster model	Li, Ma and Wang (2009) [7]
	Cluster analysis	Pardillo-Mayora et al. (2010) [8]
	Negative binomial regression	Esawey and Sayed (2012) [9]
	Cross-sectional method	Park and Abdel-Aty (2015) [10]
Fuzzy synthetic method	A set pair analysis model	Wei and Zhang (2011) [11]
	Fuzzy judgment	Fang et al. (2013) [12]
Probability theory	Evidential reasoning method	Ayati et al. (2012) [13]
	Reliability analysis	Jalayer and Zhou (2016) [14]

**Table 2 ijerph-16-01166-t002:** Band values for experts’ evaluations.

Degree of Safety Risk	$P\left(B_{i}=Hi\right)$	$P\left(B_{i}=Lo\right)$
Very significant	0.8–1.0 (0.9)	0.0–0.2 (0.1)
Significant	0.6–0.8 (0.7)	0.2–0.4 (0.3)
Potentially significant	0.4–0.6 (0.5)	0.4–0.6 (0.5)
Low significant	0.2–0.4 (0.3)	0.6–0.8 (0.7)
Very low significant	0.0–0.2 (0.1)	0.8–1.0 (0.9)

**Table 3 ijerph-16-01166-t003:** Safety risk levels on a scale of one to five.

Band Values	Safety Risk Level
0≤P(R=Hi)<0.2	1-level
0.2≤P(R=Hi)<0.4	2-level
0.4≤P(R=Hi)<0.6	3-level
0.6≤P(R=Hi)<0.8	4-level
0.8≤P(R=Hi)<1	5-level

**Table 4 ijerph-16-01166-t004:** Studies on main factors related to roadside safety risk/run-off-road (ROR) crashes.

SN	Main Factors	References
1	Horizontal curves radius	Liu and Subramanian (2009) [23], Mclaughlin et al. (2009) [16], Wei and Zhang (2011) [11], Lord et al. (2011) [2], Fang et al. (2013) [12], Eustace et al. (2014) [21], Roque et al. (2015) [22], Roque and Jalayer (2018) [24], Loprencipe et al. (2018) [6]
2	Longitudinal gradient	Liu and Subramanian (2009) [23], Li, Ma and Wang (2009) [7], Wei and Zhang (2011) [11], Fang et al. (2013) [12], Eustace et al. (2014) [21], Roque and Jalayer (2018) [24], Loprencipe et al. (2018) [6]
3	Side slope grade	Stonex (1960) [15], Zegeer et al. (1987) [4], Lee and Mannering (2002) [19], Mclaughlin et al. (2009) [16], Li, Ma and Wang (2009) [7], Pardillo-Mayora et al. (2010) [8], You et al. (2010) [5], Lord et al. (2011) [2], Roque et al. (2015) [22], Jalayer and Zhou (2016) [14]
4	Side slope height	
5	Distance between roadway edge and non-traversable obstacles	Zegeer et al. (1987) [4], Lee and Mannering (2002) [19], Sperry et al. (2008) [18], Li, Ma and Wang (2009) [7], Pardillo-Mayora et al. (2010) [8], You et al. (2010) [5], Wei and Zhang (2011) [11], Lord et al. (2011) [2], Esawey and Sayed (2012) [9], Fang et al. (2013) [12], Fitzpatrick et al. (2014) [17], Park and Abdel-Aty (2015) [10], Jalayer and Zhou (2016) [14], Roque and Jalayer (2018) [24]
6	Density of discrete non-traversable obstacles (e.g., trees, utility poles, buildings, etc.)	Stonex (1960) [15], Lee and Mannering (2002) [19], Holdridge et al. (2005) [20], Sperry et al. (2008) [18], Li, Ma and Wang (2009) [7], You et al. (2010) [5], Esawey and Sayed (2012) [9], Ayati et al. (2012) [13], Park and Abdel-Aty (2015) [10], Loprencipe et al. (2018) [6]
7	Density of continuous non-traversable obstacles (e.g., worn out roadside safety barriers, unprotected drainage channels, etc.)	Stonex (1960) [15], Holdridge et al. (2005) [20], Sperry et al. (2008) [18], Li, Ma and Wang (2009) [7], You et al. (2010) [5], Ayati et al. (2012) [13], Loprencipe et al. (2018) [6]
8	Bridge rails	Holdridge et al. (2005) [20]
9	Speed limit	Liu and Subramanian (2009) [23]
10	Lighting conditions	Liu and Subramanian (2009) [23]
11	Traffic volume	Lord et al. (2011) [2]
12	Sight distance	Wei and Zhang (2011) [11]
13	Lane width	Roque and Jalayer (2018) [24]

**Table 5 ijerph-16-01166-t005:** Decision criteria of the respective factors for three expert panels.

Factors	Expert Panel-1		Expert Panel-2		Expert Panel-3	
	Criteria	$P\left(B_{i}=Hi|E_{i1}=e_{k}\right)$	Criteria	$P\left(B_{i}=Hi|E_{i2}=e_{k}\right)$	Criteria	$P\left(B_{i}=Hi|E_{i3}=e_{k}\right)$
$B_{1}$ (m)	<30	0.5	<20	0.6	<15	0.7
	[30,60]	0.3	[20,40)	0.4	[15,30)	0.5
	>60	0.1	[40,60]	0.2	[30,45)	0.3
	-	-	>60	0.1	[45,60]	0.2
	-	-	-	-	>60	0.1
$B_{2}$ (%)	>3.0	0.5	>4.0	0.6	>3.0	0.5
	[1.0,3.0]	0.35	[2.0,4.0]	0.45	[2.0,3.0]	0.4
	<1.0	0.1	[1.0,2.0)	0.3	[1.0,2.0)	0.25
	-	-	<1.0	0.15	<1.0	0.1
$B_{3}$ (m)	<1.0	0.5	<0.5	0.6	<0.5	0.6
	[1.0,1.5]	0.3	[0.5,1.0)	0.4	[0.5,1.0)	0.45
	>1.5	0.1	[1.0,1.5]	0.2	[1.0,1.5)	0.35
	-	-	>1.5	0.1	[1.5,2.0]	0.25
	-	-	-	-	>2.0	0.1
$B_{4}$	>1:1	0.5	>1:1	0.6	>1:1	0.5
	[1:4,1:1]	0.35	[1:2,1:1]	0.45	[1:3,1:1]	0.4
	<1:4	0.15	[1:4,1:2)	0.2	[1:4,1:3)	0.25
	-	-	<1:4	0.1	<1:4	0.1
$B_{5}$ (m)	>1.5	0.4	>2.0	0.5	>3.0	0.6
	[0.5,1.5]	0.25	[1.0,2.0]	0.4	[2.0,3.0]	0.5
	<0.50	0.15	[0.5,1.0)	0.2	[1.0,2.0)	0.35
	-	-	<0.50	0.1	<1.0	0.2
$B_{6}$ (point/km)	>20	0.6	>20	0.5	>25	0.6
	[10,20]	0.45	[10,20]	0.4	[10,25]	0.45
	<10	0.2	[5,10)	0.2	[5,15)	0.25
	-	-	<5	0.1	<5	0.15
$B_{7}$ (obstacle/km)	>30	0.4	>40	0.5	>30	0.5
	[10,30]	0.2	[20,40]	0.3	(20,30]	0.35
	<10	0.1	<20	0.2	[10,20]	0.2
	-	-	-	-	<10	0.1
$B_{8}$ (km/km)	>0.2	0.4	>0.3	0.45	>0.3	0.4
	[0.1,0.2]	0.25	[0.1,0.3]	0.3	(0.2,0.3]	0.35
	<0.1	0.1	<0.1	0.15	[0.1,0.2]	0.2
	-	-	-	-	<0.1	0.1

**Table 6 ijerph-16-01166-t006:** Conditional probability $P\left(E_{3j}|B_{3}\right)$ for three expert panels.

Expert Panel	Conditional Probability Falling in Hi	Conditional Probability Falling in Lo
Expert panel-1	$P\left(E_{31}=e_{1}|B_{3}=Hi\right)=0.5556$	$P\left(E_{31}=e_{1}|B_{3}=Lo\right)=0.2381$
	$P\left(E_{31}=e_{2}|B_{3}=Hi\right)=0.3333$	$P\left(E_{31}=e_{2}|B_{3}=Lo\right)=0.3333$
	$P\left(E_{31}=e_{3}|B_{3}=Hi\right)=0.1111$	$P\left(E_{31}=e_{3}|B_{3}=Lo\right)=0.4286$
Expert panel-2	$P\left(E_{32}=e_{1}|B_{3}=Hi\right)=0.4615$	$P\left(E_{32}=e_{1}|B_{3}=Lo\right)=0.1481$
	$P\left(E_{32}=e_{2}|B_{3}=Hi\right)=0.3077$	$P\left(E_{32}=e_{2}|B_{3}=Lo\right)=0.2222$
	$P\left(E_{32}=e_{3}|B_{3}=Hi\right)=0.1538$	$P\left(E_{32}=e_{3}|B_{3}=Lo\right)=0.2963$
	$P\left(E_{32}=e_{4}|B_{3}=Hi\right)=0.0769$	$P\left(E_{32}=e_{4}|B_{3}=Lo\right)=0.3333$
Expert panel-3	$P\left(E_{33}=e_{1}|B_{3}=Hi\right)=0.3429$	$P\left(E_{33}=e_{1}|B_{3}=Lo\right)=0.1231$
	$P\left(E_{33}=e_{2}|B_{3}=Hi\right)=0.2571$	$P\left(E_{33}=e_{2}|B_{3}=Lo\right)=0.1692$
	$P\left(E_{33}=e_{3}|B_{3}=Hi\right)=0.2000$	$P\left(E_{33}=e_{3}|B_{3}=Lo\right)=0.2000$
	$P\left(E_{33}=e_{4}|B_{3}=Hi\right)=0.1429$	$P\left(E_{33}=e_{4}|B_{3}=Lo\right)=0.2308$
	$P\left(E_{33}=e_{5}|B_{3}=Hi\right)=0.0571$	$P\left(E_{33}=e_{5}|B_{3}=Lo\right)=0.2769$

**Table 7 ijerph-16-01166-t007:** Crash severity and crash frequency from 2015 to 2016 in the study area.

Category (Coefficient)		Frequency	Percentage
Crash severity	Fatal (2.0)	12	6.0%
	Injury (1.5)	76	38.2%
	Property Damage Only (PDO) (1.0)	111	55.8%
	No Crash (NC) (0)	0	0.0%
	Total	199	100%
Crash frequency (number of ROR crashes per segment)	0	8	8.1%
	1	22	22.2%
	2	35	35.4%
	3	25	25.2%
	≥4	9	9.1%
	Total	99	100%

**Table 8 ijerph-16-01166-t008:** Summary statistics for CSI corresponding to safety risk levels.

	CSI Corresponding to Levels in an Experiment Group				CSI Corresponding to Levels in a Control Group			
Levels	Mean	SD	Min.	Max.	Mean	SD	Min.	Max.
1	1.32	0.841	0.0	4.0	1.72	1.087	0.0	5.5
2	2.95	0.772	1.0	5.0	2.82	1.965	0.0	9.0
3	4.90	1.020	3.5	7.0	5.10	1.758	3.0	8.0
4	6.69	0.496	6.0	7.5	5.31	2.076	1.0	7.5
5	8.83	0.624	8.0	9.5	6.33	2.461	3.5	9.5

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
