# Peer review of "Evaluating the Safety Risk of Rural Roadsides Using a Bayesian Network Method"

_ijerph, 2019, doi:10.3390/ijerph16071166_

Round 1
Reviewer 1 Report
The article 'Evaluating the safety risk of rural roadsides using a Bayesian Network Method' proposed a Bayesian Network Based method using the experts' judgments on the conditional probability of different safety risk factors to evaluate the safety risk of rural roadsides in Nantong, China.This is a important subject to research. I suggest the authors improve the informations for future researchs and inset the limitations of the work done in Section 5. Conclusions.
Author Response
Response to Reviewer 1 Comments
We are very grateful to the reviewers for their valuable comments. We have carefully gone through the reviewers’ comments and revised the paper accordingly. All the changes made in the revised manuscript are shown in red text for convenience. Below are our detailed responses to the reviewer comments. The reviewer comments are in black text, and our responses are in red text.
Point 1: The article 'Evaluating the safety risk of rural roadsides using a Bayesian Network Method' proposed a Bayesian Network Based method using the experts' judgments on the conditional probability of different safety risk factors to evaluate the safety risk of rural roadsides in Nantong, China. This is a important subject to research. I suggest the authors improve the information for future researches and inset the limitations of the work done in Section 5. Conclusions.
Response 1: Thank you for the comment. To clarify the limitations of this study and future research, the revised paper includes the following statement in the conclusion section in page 18 line 514:
“A limitation of this study is that we can only obtain two-year (2015-2016) data from Nantong DOTP due to government restriction on traffic accident data disclosure. Additional studies are needed to include three or more years of data to validate our research once such data is available to the public.”
And in page 18 line 521:
“Another potential research direction is to include additional factors related to the roadside safety risk, such as lane width, sight distance, etc., which can potentially improve the quality of the proposed method in evaluating the safety risk of rural roadsides.”
Reviewer 2 Report
The paper has an interesting objective and addresses a topic of great interest at present. The methodology is appropriate, but it is presented as just a case study. However, it could be presented as a generic method with application to a case study as validation of use.
Another serious problem is the separation of one of the eight criteria used, both in the literature review and in the analysis of results. The eight criteria should be presented together, as a result of the bibliographical research and not as seven criteria to which one is added in the subsection 3.2. Why only this criterion is eliminated from the analysis to compare results with it and without it?
In addition, the structure of the document is deficient, since there is a lack of adequate organization that presents the safety risk factors and evaluating methods well-separated. The authors present a subsection for each of these topics in the literature review, but the contents appear mixed, since many techniques are cited in the subsection on the factors. It is recommended that these contents be separated properly. In addition, Table 1 is presented as a summary of the factors related to the objective of the study at the end of subsection 2.2 (so it is expected to summarize the methods). It is considered that this table could be completed with a summary of the methods found, and thus be presented as a result of the study. Therefore, this table could be presented at the beginning of section 4.
In addition, the case study is already presented in the introduction section very extensively and is repeated in the subsection of the study area. A shorter reference to the case study is recommended in the introduction.
Subsection 3.2 presents the criteria introduced in the method (referring to Table 1), but it does not make sense for it to be located as Materials and Methods. Again, it is recommended that it be located in the results section, but the "access point density" criterion should be presented as the other criteria in the literature review. On the other hand, the definition of “continuous non-traversable obstacles” at the end of this subsection must be justified.
The definition of the panel of experts used in the research should be presented in the Materials and Methods section, and not in the results.
Moreover, to improve the presentation of the text for publication of this study in International Journal of Environmental Research and Public Health the authors also should review the aspects described below:
· A major revision of the English language is required. For example, there are numerous mistakes in relation with coherence among nouns, verbs and articles, and incorrect terms throughout the text. There are also quite a few repetitions between different sections of the article that must be eliminated.
· The abbreviations ROR and BN are used in the abstract without previous presentation of the extended terms.
· Most studies on road safety use traffic accident data of at least 3 years to avoid biased results due to the cyclical variability of accidents. Therefore, the use of data of only 2 years in the validation of the study should be justified.
· Although the literature review is presented in section 2, there is an important lack of adequate references to related work in the introduction section, especially in the second paragraph.
· The conclusions are scarce and should present the results of the paper as a generic methodology applied to a case study, with emphasis on the criteria studied.
Author Response
Response to Reviewer 2 Comments
We are very grateful to the reviewers for their valuable comments. We have carefully gone through the reviewers’ comments and revised the paper accordingly. All the changes made in the revised manuscript are shown in red text for convenience. Below are our detailed responses to the reviewer comments. The reviewer comments are in black text, and our responses are in red text.
Point 1: The paper has an interesting objective and addresses a topic of great interest at present. The methodology is appropriate, but it is presented as just a case study. However, it could be presented as a generic method with application to a case study as validation of use.
Response 1: Thank you for the comment. We reorganized the paper and presented the proposed method in Section 3 and presented the case study in section 4 in the revised paper. Meanwhile, we revised the following statement in page 2 line 92:
“The remainder of the paper is organized as follows. In section 2, literature related to rural roadside safety are reviewed. In section 3, a safety risk assessment model for rural roadsides is developed based on a BN method. Then a case study and discussions are proposed in section 4. Finally, some conclusions are provided in the last section.”
Point 2: Another serious problem is the separation of one of the eight criteria used, both in the literature review and in the analysis of results. The eight criteria should be presented together, as a result of the bibliographical research and not as seven criteria to which one is added in the subsection 3.2. Why only this criterion is eliminated from the analysis to compare results with it and without it?
Response 2: Thank you for the comment. In the revised paper, we presented eight criteria together. We added literature related access point density in literature review section (subsection 2.1). The revised paper includes the following statement in the literature review section in page 4 line 165:
“In addition, few studies have the perceived access point density as a key factor contributing to ROR crashes. Results from previous studies show that access point density is positively correlated with crash frequency and severity [25, 26]. Therefore, this paper added access point density to the set of factors and studied its impact on rural roadside safety risk.”
Taking the advice, we relocated subsection 3.2 in the results section to subsection 4.2.
The reason to compare results with the perceived access point density and results without it is this factor is often not included in previous studies, and by conducting a with-and-without analyse, we can validate its importance in contributing to ROR crashes. To clarify this, the revised paper includes the following statement in page 16 line 465:
“To validate the importance of factoring the perceived access point density in ROR crashes (which was often not included in previous studies), we used the proposed method to compare the results with the perceived access point density and the results without it.”
Point 3: In addition, the structure of the document is deficient, since there is a lack of adequate organization that presents the safety risk factors and evaluating methods well-separated. The authors present a subsection for each of these topics in the literature review, but the contents appear mixed, since many techniques are cited in the subsection on the factors. It is recommended that these contents be separated properly. In addition, Table 1 is presented as a summary of the factors related to the objective of the study at the end of subsection 2.2 (so it is expected to summarize the methods). It is considered that this table could be completed with a summary of the methods found, and thus be presented as a result of the study. Therefore, this table could be presented at the beginning of section 4.
Response 3: Thank you for the comment. We have reorganized the structure of the literature review section in the revised paper to better present the safety risk factors and evaluating methods. Subsection 2.1 reviews literature related two main types of safety risk factor identification methods, including field study and mathematical statistical analysis. In addition, we added a paragraph to summarize the safety risk factors confirmed by evaluating roadside safety risk. Subsection 2.2 reviews literature on evaluation method for roadside safety risk according the method types, including multi-index comprehensive evaluation method,mathematical statistical analysis,fuzzy synthetic method and probability theory. To present the safety risk factors together, we deleted the safety risk factors presented in subsection 2.2, and relocated these factors to subsection 2.1 in page 4 line 161:
“Several studies developed evaluation methods based on roadside safety risk factors to identify the main factors impacting the roadside safety risk, including horizontal curve radius, side slop, side slope height, clear zone width, longitudinal grade, distance between non-traversable obstacles and roadway edge, roadside objects density, sight distance et al [4-14].”
To further clarify this, we added a summary table 1 for the evaluation methods in subsection 2.2 and relocated a summary table 4 of the safety risk factors at subsection 4.2.
Point 4: In addition, the case study is already presented in the introduction section very extensively and is repeated in the subsection of the study area. A shorter reference to the case study is recommended in the introduction.
Response 4: Thank you for the comment. We presented the case study in a condensed version in introduction section of the revised paper in page 2 line 78:
“To validate the effectiveness of the proposed methods, a case study is conducted using a rural area (6 km2) with 19.42 km roads in Nantong, China. Nantong is a prefecture-level city in Eastern China, and its road system contains mainly rural roads (i.e., 80.5% of its 14694 km roads is classified as rural roads) [3].”
Point 5: Subsection 3.2 presents the criteria introduced in the method (referring to Table 1), but it does not make sense for it to be located as Materials and Methods. Again, it is recommended that it be located in the results section, but the "access point density" criterion should be presented as the other criteria in the literature review. On the other hand, the definition of “continuous non-traversable obstacles” at the end of this subsection must be justified.
Response 5: Thank you for the comment. According to the suggestion, subsection 3.2 relocated in the case study section in subsection 4.2. Meanwhile, we revised some statements about the number of criteria in methods section in page 8 line 281 and page 9 line 331, 333.
Taking the advice, we presented the "access point density" criterion as the other criteria in the literature review, the revised paper includes the following statement in the literature review section in page 4 line 165:
“In addition, few studies have the perceived access point density as a key factor contributing to ROR crashes. Results from previous studies show that access point density is positively correlated with crash frequency and severity [25, 26]. Therefore, this paper added access point density to the set of factors and studied its impact on rural roadside safety risk.”
To justify the definition of “continuous non-traversable obstacles”, we included additional references related to the definition of “continuous non-traversable obstacles” in page 11 line 375.
Point 6: The definition of the panel of experts used in the research should be presented in the Materials and Methods section, and not in the results.
Response 6: Thank you for the comment. The definition of the panel of experts are related to the chosen safety risk factors. Hence, we think that it is reasonable to present the definition after the discussion on the safety risk factors (i.e., subsection 4.2). So we put the definition in the case study (i.e., subsection 4.3).
Point 7: A major revision of the English language is required. For example, there are numerous mistakes in relation with coherence among nouns, verbs and articles, and incorrect terms throughout the text. There are also quite a few repetitions between different sections of the article that must be eliminated.
Response 7: Thank you for the comment. We have checked the English writing carefully, including problems related to terminology、typographical errors、word inconsistent、tense etc., and deleted some repetitions between different sections. Some repetitions that were deleted are in page 2 line 80, page 8 line 281, page 17 line 501 and page 18 line 511.
Point 8: The abbreviations ROR and BN are used in the abstract without previous presentation of the extended terms.
Response 8: Thank you for the comment. Taking the advice, we added the abbreviation BN and the extended term run-off-road in the abstract in page 1 line 15, 21.
Point 9: Most studies on road safety use traffic accident data of at least 3 years to avoid biased results due to the cyclical variability of accidents. Therefore, the use of data of only 2 years in the validation of the study should be justified.
Response 9: Thank you for the comment. We agree that many studies use the traffic accident data of at least 3 years to avoid potential biased results due to the cyclical variability of accidents, and also can better validate the effectiveness of evaluating method and results. The reason for using data of 2 years is that there are restrictions on the disclosure of traffic accident data in China, we can only obtain the data of 2015-2016 from the Nantong Department of Traffic Police. To address this limitation, we added the limitations of this study in the conclusion section in page 18 line 514:
“A limitation of this study is that we can only obtain two-year (2015-2016) data from Nantong DOTP due to government restriction on traffic accident data disclosure. Additional studies are needed to include three or more years of data to validate our research once such data is available to the public.”
Point 10: Although the literature review is presented in section 2, there is an important lack of adequate references to related work in the introduction section, especially in the second paragraph.
Response 10: Thank you for the comment. We included additional references in the introduction section in page 2 line 56, 57.
Point 11: The conclusions are scarce and should present the results of the paper as a generic methodology applied to a case study, with emphasis on the criteria studied.
Response 11: Thank you for the comment. We added and revised the following statements in conclusions section in page 17 line 492:
“Previous studies rely heavily on the quality of ROR crash data to evaluate rural road safety risk, but such data may not be available or complement in many rural areas due to lack of funding or manmade errors. This study proposed a BN-based method using the experts’ judgments on the conditional probability of different safety risk factors to evaluate the safety risk of rural roadsides. By summarizing the roadside safety risk factors identified in previous studies, and introducing a new factor which may have a significantly impact on the roadside safety risk, eight factors were considered, including horizontal curves radius, longitudinal gradient, side slope grade, side slope height, distance between roadway edge and non-traversable obstacles, access point density, density of discrete non-traversable obstacles and density of continuous non-traversable obstacles.
To validate the proposed method, a case study was conducted using a rural area (6 km2) with 19.42 km roads in Nantong, China. By comparing the safety risk levels of the proposed method and CSI generated from ROR crash data from 2015-2016, the results show that the proposed method can generate a consistent result with ROR crash data, meaning the higher the safety risk levels, the higher the CSI. Additionally, by comparing the evaluating results with the perceived access point density and the results without it, we also found that access point density significantly contributed to the safety risk of rural roadsides. These results demonstrate that the proposed method can cope with inconsistent expert judgments, and can serve as a low-cost solution to evaluate the safety risk of rural roadsides, especially for areas with incomplete ROR crash data.”
Reviewer 3 Report
Dear Authors,
here is my response to the manuscript entitled “Evaluating the safety risk of rural roadsides using a Bayesian Network Method”. My suggested revisions are rather minor, so after revision I recommend publication.
Comments on abstract, title, references
The title is informative and relevant. The methodology is outlined and the conclusions seem to align with the aim. It is clear what the study found and how the Authors did it, nevertheless the aim of the research is not clear in the comparison with the multiple results presented.
The references cited by the Authors are mostly up-to-date but, in my opinion, there are some crucial references missing especially in the field of Bayesian networks (Norman Fenton, Martin Neil).
Comments on introduction/background
The context of the research is clearly established and the literature review seem to be sufficient. Extensive and it seems that an optimal literature review in the field of safety risk of rural roadsides. Much less information about the application of Bayesian Networks in this research area – this gap should be filled.
Line 47 – “some recent studies proposed different methods to evaluate the rural roadside safety risk by analyzing a variety of ROR crash data.” – what are these studies?? The authors do not refer to any references confirming that statement although they suggest there are some studies on the subject (methods and different approaches are described later, but the references should be present when this kind of information is mentioned).
Comments on methodology
The process of subject selection is clear. The study methods are valid and reliable, as the information about the experts (the number of them, area of their expertise) has been provided. It seems that the description contain enough detail in order to replicate the study.
Author Response
Response to Reviewer 3 Comments
We are very grateful to the reviewers for their valuable comments. We have carefully gone through the reviewers’ comments and revised the paper accordingly. All the changes made in the revised manuscript are shown in red text for convenience. Below are our detailed responses to the reviewer comments. The reviewer comments are in black text, and our responses are in red text.
Point 1: Comments on abstract, title, references: The title is informative and relevant. The methodology is outlined and the conclusions seem to align with the aim. It is clear what the study found and how the Authors did it, nevertheless the aim of the research is not clear in the comparison with the multiple results presented.
Response 1: Thank you for the comment. To clarify the objective of this study, we rewrite some parts of the abstract as follows:
“Evaluating the safety risk of rural roadsides is critical to achieve reasonable allocation of limited budget and avoid excessive installation of safety facilities. To assess the safety risk of rural roadsides when the crash data is unavailable or missing, this study proposed a Bayesian Network (BN) method that uses the experts’ judgments on the conditional probability of different safety risk factors to evaluate the safety risk of rural roadsides. Eight factors were considered, including seven factors identified in the literature and a new factor named access point density. To validate the effectiveness of the proposed method, a case study was conducted using 19.42-kilometer-long road networks in the rural area of Nantong, China. By comparing the results of the proposed method and run-off-road (ROR) crash data from 2015-2016 in the study area, the road segments with higher safety risk levels identified by the proposed method are found to be statistically significantly correlate with higher crash severity based on the crash data. In addition, by comparing the respective results evaluated by eight factors and seven factors (a new factor removed), we also found that access point density significantly contributed to the safety risk of rural roadsides. These results show that the proposed method can be considered as a low-cost solution to evaluate the safety risk of rural roadsides with relatively high accuracy, especially for areas with large rural road networks and incomplete ROR crash data due to budget limitation, human errors, negligence or inconsistent crash recordings.”
Point 2: The references cited by the Authors are mostly up-to-date but, in my opinion, there are some crucial references missing especially in the field of Bayesian networks (Norman Fenton, Martin Neil).
Response 2: Thank you for the comment. We included additional references related to Bayesian networks in section 3 in page 7 line 275.
Point 3: Comments on introduction/background: The context of the research is clearly established and the literature review seem to be sufficient. Extensive and it seems that an optimal literature review in the field of safety risk of rural roadsides. Much less information about the application of Bayesian Networks in this research area – this gap should be filled.
Response 3: Thank you for the comment. The revised paper includes the following statement in literature review section in page 5 line 209:
“Apart from the aforementioned methods, Bayesian Network has been used in traffic safety evaluation domain. Several studies used BN to assess traffic safety, traffic accidents and traffic accident injury severity [27-29]. Some other studies identified the accident black spots and predicted real-time crash using a BN method [30,31]. However, few studies used BN to assess rural roadside safety risk.”
Point 4: Line 47 – “some recent studies proposed different methods to evaluate the rural roadside safety risk by analyzing a variety of ROR crash data.” – what are these studies?? The authors do not refer to any references confirming that statement although they suggest there are some studies on the subject (methods and different approaches are described later, but the references should be present when this kind of information is mentioned).
Response 4: Thank you for the comment. Taking the advice, we included additional references related to the claims we made in the introduction section in page 2 line 57.
Point 5: Comments on methodology: The process of subject selection is clear. The study methods are valid and reliable, as the information about the experts (the number of them, area of their expertise) has been provided. It seems that the description contain enough detail in order to replicate the study.
Response 5: Thank you for the comment.
Reviewer 4 Report
The article contains information technical and innovative that justifies its publication. The problem addressed is current and has technical relevance, which makes it significant. The abstract is written concisely. The paper is well organized and convincing. The experimental methodology is described comprehensively. Interpretations and conclusions are justified by the results. The paper is well organized and convincing.
My recommendations are:
* What is the motivation of the proposed work? Research gaps, objectives of the proposed work should be clearly justified;
* Bullet your contribution at the end of the introduction section;
* Introduction section can be extended to add the issues in the context of the existing work and how proposed approach can be used to overcome this;
* Clarify the finding Error rate and accuracy in performance analysis section;
* Authors should add more details about the implementation of the code to perform the analysis and the library involved in this task;
* Quality of Figures is so important too. Please provide some high-resolution figures. The comparison of different methods using clear graphs should be explained;
* Please add information about the time to label a new sample under analysis. How does this work compare to other works? The contributions of this work need to be clearly articulated. The author might consider justifying the performance of this study with recent study and methods.
Author Response
Response to Reviewer 4 Comments
We are very grateful to the reviewers for their valuable comments. We have carefully gone through the reviewers’ comments and revised the paper accordingly. All the changes made in the revised manuscript are shown in red text for convenience. Below are our detailed responses to the reviewer comments. The reviewer comments are in black text, and our responses are in red text.
Point 1: The article contains information technical and innovative that justifies its publication. The problem addressed is current and has technical relevance, which makes it significant. The abstract is written concisely. The paper is well organized and convincing. The experimental methodology is described comprehensively. Interpretations and conclusions are justified by the results. The paper is well organized and convincing.
Response 1: Thank you for the comment.
Point 2: What is the motivation of the proposed work? Research gaps, objectives of the proposed work should be clearly justified
Response 2: Thank you for the comment. The motivation of the proposed work is to evaluate and classify the safety risk levels of rural roadsides to prioritize budget allocation and possible safety actions. Meanwhile, the methodology in the previous studies are entirely dependent on the data quality of ROR crash. In reality, many types of ROR crash data are not available or missing in many countries. The study proposes a BN-based method to address this issue.
To clarify the objective of this study, we rewrite some parts of the abstract as follows:
“Evaluating the safety risk of rural roadsides is critical to achieve reasonable allocation of limited budget and avoid excessive installation of safety facilities. To assess the safety risk of rural roadsides when the crash data is unavailable or missing, this study proposed a Bayesian Network (BN) method that uses the experts’ judgments on the conditional probability of different safety risk factors to evaluate the safety risk of rural roadsides. Eight factors were considered, including seven factors identified in the literature and a new factor named access point density. To validate the effectiveness of the proposed method, a case study was conducted using 19.42-kilometer-long road networks in the rural area of Nantong, China. By comparing the results of the proposed method and run-off-road (ROR) crash data from 2015-2016 in the study area, the road segments with higher safety risk levels identified by the proposed method are found to be statistically significantly correlate with higher crash severity based on the crash data. In addition, by comparing the respective results evaluated by eight factors and seven factors (a new factor removed), we also found that access point density significantly contributed to the safety risk of rural roadsides. These results show that the proposed method can be considered as a low-cost solution to evaluate the safety risk of rural roadsides with relatively high accuracy, especially for areas with large rural road networks and incomplete ROR crash data due to budget limitation, human errors, negligence or inconsistent crash recordings.”
And in the introduction section in page 2 line 66:
“To evaluate the safety risk of rural roadsides for areas with incomplete ROR crash data, this paper proposes a Bayesian Network-based method to evaluate the rural roadside safety risk based on the estimated probability of ROR crash using expert’s judgment.”
Point 3: Bullet your contribution at the end of the introduction section.
Response 3: Thank you for the comment. The revised paper includes the following statements in the introduction section in page 2 line 84:
“(1) A Bayesian Network-based method to evaluate the safety risk of rural roadsides for areas with incomplete ROR crash data is proposed.
(2) Access point density is introduced as a relatively new risk factor to evaluate the safety risk of rural roadsides, and the results demonstrated that it has a significant impact on the safety risk of rural roadsides.
(3) This proposed method can be applied to assess the safety risk of road segments or intersections in the absence of ROR crash data as long as safety risk factors for road segments or intersections are re-identified.”
Point 4: Introduction section can be extended to add the issues in the context of the existing work and how proposed approach can be used to overcome this.
Response 4: Thank you for the comment. There two issues in the context of the existing work. One issue is that the effectiveness and practicality of previous methodology are entirely dependent on the data quality of ROR crash. The other issue is that safety risk factors for rural roadsides considered in the previous studies are not comprehensive, which can not accurately characterize the rural roadside feature, thus the relationship obtained in the previous studies between the factors and safety risk is also biased. This paper proposes a Bayesian Network-based method to overcome the issue of incomplete ROR crash data. Then, based on the frequency and importance of safety risk factors identified in previous study, and introducing a new factor called access point density, we defined a set of safety risk factors to provide a relatively complete picture of rural roadside safety risk. To clarify this, the revised paper includes the following statement in the introduction section in page 2 line 62:
“In addition, some of the safety risk factors for rural roadsides were not included in previous studies, which may not provide a complete picture of rural roadside safety risk. Thus the relationship obtained in previous studies between the factors and safety risk is may also biased [15-24].”
In addition, the following statements were added in page 2 line 69:
“To provide a relatively complete picture of rural roadside safety risk, seven safety risk factors were considered based on their frequency and importance in the literature. A new factor named access point density is introduced to capture some roadside features that was not included in previous studies and may has a significantly impact on the frequency and severity of ROR crashes.”
Point 5: Clarify the finding Error rate and accuracy in performance analysis section.
Response 5: Thank you for the comment. This paper validates the effectiveness of evaluating method and results by comparing each segment’s CSI and P(R=Hi). We used the percentage of 99 segments whose CSI and P(R=Hi) does not fall into the same risk level to clarify the error rate of the evaluating results, which is 12.1% in our case. We used a correlation coefficient (R2=0.8538) between the safety risk levels and CSI to demonstrate the accuracy of the evaluating results. The above statement can been read in page 15 line 454, 458.
Point 6: Authors should add more details about the implementation of the code to perform the analysis and the library involved in this task.
Response 6: Thank you for the comment. Because the whole road network has 99 segments in study area, the calculation scale is small. We completed all calculations efficiently using BN tools GeNie2.0. To clarify this, the following statements were added in page 13 line 412, 415:
“The above calculation results were calculated by BN tools GeNie2.0.”
“Figure 4 shows the roadside safety risk evaluation results calculated by GeNie2.0.”
Point 7: Quality of Figures is so important too. Please provide some high-resolution figures. The comparison of different methods using clear graphs should be explained.
Response 7: Thank you for the comment. We improved the quality figures in the revised paper.
Point 8: Please add information about the time to label a new sample under analysis. How does this work compare to other works? The contributions of this work need to be clearly articulated. The author might consider justifying the performance of this study with recent study and methods.
Response 8: Thank you for the comment. The evaluating results were calculated by BN tools GeNie2.0. Since the scale of the Bayesian Network is small, there is no significant difference in the calculation time when adding or removing a factor. Hence, the information of calculation time is not presented in the paper.
We agree that it is necessary to justify the performance of this study with recent study and methods. In this paper, we proposed a relatively new method to evaluate the rural road safety risk which is very different from traditional methods. Previous methods rely heavily on the quality of the ROR crash data to evaluate rural road safety risk, while we use expert judgement to conduct evaluation. Therefore, we cannot compare our results with methods used in other studies.
To clarify the differences between our method and previous methods, the following statements were added in the introduction section in page 2 line 66:
“To evaluate the safety risk of rural roadsides for areas with incomplete ROR crash data, this paper proposes a Bayesian Network-based method to evaluate the rural roadside safety risk based on the estimated probability of ROR crash using expert’s judgment.”
And in the conclusions section in page 17 line 492:
“Previous studies rely heavily on the quality of ROR crash data to evaluate rural road safety risk, but such data may not be available or complement in many rural areas due to lack of funding or manmade errors.”
Round 2
Reviewer 2 Report
Most of the suggested changes by this referee were implemented adequately in the new version of the paper. However, for the separation between the results of the article (the proposed method) and the validation in the case study to be completely evident, section 4 must separately present, on the one hand, the set of safety risk factors for rural roadsides (which do not depend of the case studied) and the structure of the method itself, and, on the other hand, the data and analysis of the case study.
With these minor changes, I recommend the publication of this manuscript in International Journal of Environmental Research and Public Health.
Author Response
Point 1: Most of the suggested changes by this referee were implemented adequately in the new version of the paper. However, for the separation between the results of the article (the proposed method) and the validation in the case study to be completely evident, section 4 must separately present, on the one hand, the set of safety risk factors for rural roadsides (which do not depend of the case studied) and the structure of the method itself, and, on the other hand, the data and analysis of the case study. With these minor changes, I recommend the publication of this manuscript in International Journal of Environmental Research and Public Health
Response 1: Thank you for the comment. To evidently separate between the proposed method and the case study, we reorganized the paper and presented the safety risk factors for rural roadsides in Section 3 in the revised paper. Meanwhile, we revised the following statement in page 2 line 79:
“The remainder of the paper is organized as follows. In section 2, literature related to rural roadside safety are reviewed. In section 3, a safety risk evaluation method for rural roadsides is developed based on a BN method and the safety risk factors are discussed. Then a case study and discussions are proposed in section 4. Finally, some conclusions are provided in the last section.”